# Myoblast-Derived Galectin 3 Impairs the Early Phases of Osteogenesis Affecting Notch and Akt Activity

**DOI:** 10.3390/biom14101243

**Published:** 2024-09-30

**Authors:** Emanuela Amore, Vittoria Cenni, Manuela Piazzi, Michele Signore, Giulia Orlandi, Simona Neri, Stefano Biressi, Rosario Barone, Valentina Di Felice, Matilde Y. Follo, Jessika Bertacchini, Carla Palumbo

**Affiliations:** 1Laboratorio Ramses, IRCCS Istituto Ortopedico Rizzoli, Via di Barbiano 1/10, 40136 Bologna, Italy; emanuela.amore@ior.it; 2CNR-Institute of Molecular Genetics, 40136 Bologna, Italy; vittoria.cenni@cnr.it (V.C.); manuela.piazzi@cnr.it (M.P.); 3IRCCS Istituto Ortopedico Rizzoli, 40136 Bologna, Italy; 4RPPA Unit of Proteomics Area, Core Facilities, Istituto Superiore di Sanità, 00161 Rome, Italy; michele.signore@iss.it; 5Department of Surgery, Medicine, Dentistry and Morphological Sciences with Interest in Transplant, Oncology and Regenerative Medicine, University of Modena and Reggio Emilia, 41124 Modena, Italy; giulia.orlandi90@unimore.it; 6Medicine and Rheumatology Unit, IRCCS Istituto Ortopedico Rizzoli, 40136 Bologna, Italy; simona.neri@ior.it; 7Department of Cellular, Computational and Integrative Biology, University of Trento, 38123 Trento, Italy; stefano.biressi@unitn.it; 8Department of Biomedicine, Neuroscience and Advanced Diagnostics, University of Palermo, 90133 Palermo, Italy; valentina.difelice@unipa.it; 9Department of Biomedical and Neuromotor Sciences, University of Bologna, 40127 Bologna, Italy; matilde.follo@unibo.it; 10Department of Biomedical, Metabolic and Neural Sciences, Section of Human Morphology, University of Modena and Reggio Emilia, 41124 Modena, Italy; carla.palumbo@unimore.it

**Keywords:** galectins, Akt, Notch, muscle-to-bone crosstalk, myokine, proteomics

## Abstract

Galectin-3 (Gal-3) is a pleiotropic lectin produced by most cell types, which regulates multiple cellular processes in various tissues. In bone, depending on its cellular localization, Gal-3 has a dual and opposite role. If, on the one hand, intracellular Gal-3 promotes bone formation, on the other, its circulating form affects bone remodeling, antagonizing osteoblast differentiation and increasing osteoclast activity. From an analysis of the secretome of cultured differentiating myoblasts, we interestingly found the presence of Gal-3. After that, we confirmed that Gal-3 was expressed and released in the extracellular environment from myoblast cells during their differentiation into myotubes, as well as after mechanical strain. An in vivo analysis revealed that Gal-3 was triggered by trained exercise and was specifically produced by fast muscle fibers. Speculating a role for this peptide in the muscle-to-bone cross talk, a direct co-culture in vitro system, simultaneously combining media that were obtained from differentiated myoblasts and osteoblast cells, confirmed that Gal-3 is a mediator of osteoblast differentiation. Molecular and proteomic analyses revealed that the secreted Gal-3 modulated the biochemical processes occurring in the early phases of bone formation, in particular impairing the activity of the STAT3 and PDK1/Akt signaling pathways and, at the same time, triggering that one of Notch. Circulating Gal-3 also affected the expression of the most common factors involved in osteogenetic processes, including BMP-2, -6, and -7. Intriguingly, Gal-3 was able to interfere with the ability of differentiating osteoblasts to interact with the components of the extracellular bone matrix, a crucial condition required for a proper osteoblast differentiation. All in all, our evidence lays the foundation for further studies to present this lectin as a novel myokine involved in muscle-to-bone crosstalk.

## 1. Introduction

In adults, bone formation is a physiological process in which the resorption and the production of bone matrix co-exist (i.e., bone remodeling) within a finely tuned homeostatic balance. Properly coordinating these events is critical to maintaining the integrity and the health of bone during growth and over the whole life of individuals, or in response to injuries and mechanical loads, and, last but not least, to contribute to systemic mineral homeostasis [1]. From a molecular point of view, the differentiation of bone-forming cells (i.e., osteoblasts) occurs through the involvement of several signaling cascades, including the Akt/mTOR pathway and the molecular signaling downstream of Notch, Wnt, TGF-β, MAPKs, and the BMPs protein family [2,3,4,5,6]. All these cascades culminate in cell cycle arrest and in the activation of bone-specific transcription factors that lead to the activity of both osteoblasts and osteoclasts, resulting in the production and the resorption of the bone matrix. The homeostasis of bone cells is regulated by factors whose release is physiologically controlled during the normal growth of bone or is stimulated as a consequence of injuries or physical activity, and that can act in a paracrine or autocrine manner. Between these factors, there are several specific molecules that, depending on the circumstances, are released in circulation by nearby tissues. Specifically, skeletal muscle controls bone growth by the finely-tuned secretion of a variety of substances, including myostatin, osteoglycin, decorin, irisin, IL-6 [7,8], and, as recently published by us, sclerostin, whose effects result in an up- or down-regulation of bone growth [9]. Since the discovery of novel myokines involved in bone homeostasis may have important socio-economic implications, related, for example, to particular classes of bone disorders, but also to the increase in the average age of the population, we decided to focus our attention on the identification of the molecules secreted from muscle cells and acting on bone physiology. To this end, we intriguingly found that Gal-3 is released from muscle cells and decided to explore the role of this molecule in the interplay between muscle and bone. In particular, we found that Gal-3 is highly expressed by fast muscle fibers, where it could act as a metabolic regulatory factor during exercise-induced muscle plasticity. Galectins, also known as galactose-binding proteins, are small molecules that bind the β-galactoside residues of proteins via a carbohydrate recognition domain [10]. To date, 15 types of galectins have been identified inside the cells, either in the cytosol or in the nucleus, but some of them are also released in the extracellular matrix and the blood [11]. By binding to several types of proteins outside the cells, galectins contribute to cell–cell and cell–matrix interactions and trigger downstream molecular signaling pathways that modulate a variety of cellular functions, including immune regulation and inflammation [12]. Furthermore, intracellular galectins may interact with ligands to regulate cellular activities and contribute to some fundamental processes, such as pre-mRNA splicing. Like other galectins, Gal-3 [Gal-3], which is a 29- to 35-kDa protein expressed in various tissues, is involved in several intra- and extracellular functions, such as cell proliferation and differentiation. Gal-3 is also secreted, thereby acting in a paracrine and an autocrine manner [13]. The pleiotropic nature of this lectin makes it a protein involved in several physiological and pathological cellular contexts, including inflammation, fibrosis, and cancer [14,15,16,17]. It has been demonstrated that circulating Gal-3 can also derive from cardiac cells in response to specific cardiac pathological conditions [18]. In normal osteoblasts, several studies have shown that Gal-3 has a dual activity, depending on its cellular localization: while intracellular Gal-3 promotes osteoprogenitors’ differentiation into osteoblasts [19,20], the soluble form of Gal-3 antagonizes osteoblast differentiation by both increasing osteoclast activity and reducing osteoblast differentiation [21]. A number of studies have also described a role of Gal-3 in bone tumors. According to this evidence, Gal-3 silencing represses osteosarcoma progression in osteosarcoma cell lines, as well as in in vivo murine models [22]. Other studies have revealed that circulating Gal-3 that is released from metastatic cells promotes osteoclast differentiation [22,23]; nonetheless, other studies demonstrate that a secreted form of Gal-3, released from breast cancer cells, blocks the differentiation of bone metastasis through the deregulation of Notch signaling [21]. Therefore, understanding the regulation of bone remodeling by Gal-3 is undoubtedly a clinical priority and a challenge in the struggle against any bone disease. In the following paragraphs, we show our findings, suggesting that Gal-3 is released from myoblasts during muscle differentiation and upon mechanical stimulation. We also thoroughly describe the molecular mechanisms underlying the role of extracellular Gal-3 in modulating the early phases of osteogenesis.

## 2. Materials and Methods

### 2.1. Cell Cultures and Treatments

Cell cultures: A murine, muscle-derived C2C12 cell line was grown in a DMEM-High Glucose (GIBCO, Thermo Fisher Scientific Inc., Waltham, MA, USA) and 10% Fetal Bovine Serum (FBS, GIBCO). Muscle differentiation was induced by replacing the medium with a differentiation medium (DM), namely DMEM-HG, plus 1 µg/mL insulin (Merck Sigma-Aldrich, Milano, Italy). The DM was changed every two days. The MC-3T3 murine osteoblast cell line was grown in α-MEM and 10% FBS. Osteoblast differentiation was induced with a differentiating medium, consisting of α-MEM, 10% FBS, 0.3 mM ascorbic acid, and 4 mM b-glycerophosphate (Corning, Glendale, AZ, USA). To condition the differentiation program, the DM was supplemented by 1/3 with a culture medium from the C2C12 cells, or with recombinant Gal-3 (Cusabio Technology LLC, Houston, TX, USA) at a final concentration of 3 µg/mL.

Transfection: A mouse FLAG-tagged Gal-3 encoding plasmid (Origene, Rockville, ML, USA) was introduced to the cells using a Fugene 6 reagent (Promega, Madison, WI, USA), according to manufacturer protocol, using a ratio of 3:1. The transfected cells were harvested after 48 h.

Mechanical Strain: The cells were seeded on Collagen I-coated Bioflex culture plates (Flexcell international, Burlington, NC, USA), grown until 60% of confluence or let differentiate for two days in the DM. A multiaxial mechanical load was applied for 6 and 24 h (20% elongation at 1 Hz) with a Flexcell FX-4000T Tension System (Flexcell International) at 37 °C, 5% CO_2_. As a control, some cells were left untreated in the same incubator. At the end of the strain, the cells were lysed in an SDS-lysis buffer or with a Tri-Reagent (Merck Sigma-Aldrich, Milano, Italy) and subjected to a western blot analysis or an RNA analysis.

### 2.2. Mice Endurance Training

Sedentary and trained mice were assigned to two groups, indicated as Sed and Tr, respectively. The Tr mice were subjected to a running endurance training protocol for rodents, previously described in [24]. After the training, their tibialis anterior (TA) and gastrocnemius muscles were excised, fixed, and embedded as described by Barone and colleagues [24]. The embedded muscles were sliced into sections (5 µm) that were mounted on glass slides. Serial sections were incubated in an “antigen unmasking solution” (10 mM TRIS-EDTA, 0.05% Tween-20) for 10 min at 75 °C. Then, the sections were incubated in a humidified chamber overnight at 4 °C with monoclonal mouse anti-Galectin-3 (13 µg/mL, ab2785 Abcam, Cambridge, UK). The following day, the sections were rinsed and sequentially incubated at RT for 30 min with Horseradish Peroxidase-conjugated secondary antibodies. The colorimetric reactions were developed using 3,3′-Diaminobenzidine (DAB) (HistoLine, Milan, Italy), counterstained with hematoxylin, and mounted with glycerol jelly. Negative control sections were performed. Images were captured with an Eclipse 90i microscope (Nikon Instruments Europe BV, Nikon Instruments, Amstelveen, The Netherlands).

The IHC pictures were quantified using ImageJ software. We chose three separate photo fields at a magnification of 20× for the red and white portions of the TA and gastrocnemius muscles of correspondingly *n* = 3 animals that were grouped into Sed and Tr mice. Next, we used the toolbar icons to convert the image to 8-bit, and we quantified the “mean grey value” for each selected area. The average of the mean grey values was used to express the data, and Student’s t-test was used to perform the statistical analysis.

All the animal experiments were performed before the entry into force of Decree Law n. 26/2014, following the application of European Directive 2010/63/Eu.

### 2.3. RT-PCR

A Total RNA extraction was performed by using the PureLink™ RNA Micro Kit (Invitrogen, Waltham, MA, USA), according to manufacturer’s instructions. The Total RNA (1 μg) was reverse transcribed to cDNA using the QuantiTect Reverse Transcription Kit (Qiagen, Hilden, Germany). The cDNA samples were amplified using the Sens Mix SYBR Master Mix (Origene, Rockville, MD, USA). The primers list was inserted in Appendix A.

The experiments were carried out in triplicate for each data point. A relative gene expression quantification was performed using the comparative threshold (Ct) method (ΔΔCt), in which the relative gene expression level equals 2^−ΔΔCt^. The obtained fold changes in gene expression were normalized to GAPDH. For all the tested groups, the statistical significance was set at *p* < 0.05.

### 2.4. Protein Assays

Protein extracts and immunoblot: The TA muscles from the Sed and Tr mice was excised and frozen with three consecutive cycles of freeze–thawing. The muscles were then pulverized with liquid nitrogen, added to 100 µL of lysis buffer [25], and homogenized using an insulin syringe. The lysates were maintained in agitation O/N for 16 h and centrifuged at 14,000 rpm for 20 min at 4 °C. After protein quantification was performed, the lysates were analyzed by SDS-PAGE and the expression of Gal-3 was evaluated by a western blot analysis.

The osteoblast and myoblast cells were lysed in an SDS-lysis buffer. The total lysates or culture media were resolved by SDS-PAGE (precast from Thermo-Fisher, Thermo Fisher Scientific Inc., Waltham, MA, USA), electro-transferred onto nitrocellulose membranes (Santa Cruz Biotechnology, DBA Italia SRL, Segrate, Italy), and immunoblotted with the antibodies indicated in Appendix A.

Reverse-Phase Protein Array (RPPA): The cells were lysed in an RIPA-like lysis buffer (LB) composed of TPER (Thermo-Fisher Scientific, Waltham, MA, USA), which was supplemented with 300 mM NaCl and protease and phosphatase inhibitor cocktails (Merck Sigma-Aldrich, Milano, Italy). Next, the total protein content was quantified. The RPPA samples were prepared as previously described [26]. Alkaline phosphatase assay**:** The evaluation of the alkaline phosphatase activity of the differentiated osteoblasts treated with Gal-3 or vehicle was performed with the Alkaline Phosphatase Assay Kit (from Abcam, Cambridge, UK). At the indicated time points, the cells were lysed and processed according to the manufacturer’s protocol. At the end of the procedures, the samples were read at OD 405 nm on a microplate reader.

Alizarin red staining: The differentiated MC-3T3 cells were fixed with 4% paraformaldehyde for 20’ under constant agitation. After washes with 1× PBS, alizarin dye (Merck Sigma-Aldrich, Milano, Italy) was added for 5’ at RT under stirring. The level of mineralization was observed under the microscope and representative images were captured. To quantify the mineralization, 10% cetylpyridinium chloride was added into each well for 20 min to elute alizarin red S staining. The evaluations were obtained by a spectrophotometric reading at 570 nm. Bone metabolism array: The level of expression of the proteins involved in the differentiation of both osteoblasts and osteoclasts was evaluated through a Human Bone Metabolism Antibody Array (Abcam, Cambridge, UK). The experiment was performed following the manufacturer’s protocol. At the end of the procedures, the resulting complexes were read by the Axon GenePix laser scanner.

Extracellular matrix assay: The adhesion capacity of the cells towards specific components of the extracellular matrix was measured by the extracellular matrix (ECM) Cell Adhesion Array Kit (Merck Sigma-Aldrich, Milano, Italy). The cells were detached in a non-enzymatic way, using 5 mM of EDTA, counted, and plated on wells that specifically contained ECM ligands. The cellular suspension was allowed to attach to the surface for 16 h. At the end of the incubations, the non-adherent cells were washed out, while the adherent ones were stained and quantified by a 570 nm reading.

### 2.5. Mass Spectrometry Analysis

Secretomes: The culture media from the differentiating cells were concentrated, quantified, and added to Laemmli sample buffer. For the protein separation, equivalent amounts of extracts were resolved by SDS-PAGE on Mini-PROTEAN^®^ TGXTM 4–15% Precast gels (Bio-Rad Laboratories, CA, USA); the resulting gels were stained with Coomassie Brilliant Blue, and the bands of interest were excised, processed, and analyzed in a QExactive Hybrid Quadrupole-Orbitrap Mass Spectrometer (Thermo Scientific, Waltham, MA, USA), coupled online with a UHPL ultimate 3000 system (refer to the Appendix A for the detailed protocols and data analysis). All the proteins that were identified are listed in Appendix A. More information of Mass spectrometric analysis can be found in Appendix A.

Cell lysates: The proteins from each sample were precipitated with four times their volume of 100% cold acetone and mixed gently. The samples were centrifuged at 4000× *g* for 15 min. The supernatant was removed, and the pellet was washed with 1 mL of 70% ice-cold acetone. After being air-dried, the pellets were dissolved in 50 µL of 6M urea/200 mM ammonium bicarbonate and digested with trypsin (Thermo Fisher) at 37 °C for 4 h. The samples were reduced with DTT and alkylated in the dark with IAA. Trypsin was added, in the ratio 1:100. The digested protein samples were purified by C18-SD columns (Supelco Empore^TM^, Sigma Aldrich, Milan, Italy) and analyzed in a QExactive Hybrid Quadrupole-Orbitrap Mass Spectrometer (Thermo Scientific, Waltham, MA, USA), coupled online with a UHPL ultimate 3000 system (refer to the Appendix A for the detailed protocols and data analysis). The mass spectrometry proteomics data were deposited in the ProteomeXchange Consortium via the PRIDE [27] partner repository with the dataset identifiers PXD040187 and 10.6019/PXD040187.

### 2.6. Image Processing and Statistical Analysis

All the images shown are representative of independent experiments carried out under the same conditions. The images from the biochemical and functional studies were processed using Photoshop CS4 (Adobe Systems, Inc., San Jose, CA, USA). Densitometric analyses were performed by ImageJ (National Institute of Health, Bethesda, MD, USA). The data are expressed as the mean ± SD of the number of the indicated biological replicates. The data were analyzed by Student’s *t*-test, with a significance level of * *p* < 0.05, ** *p* < 0.01, and *** *p* < 0.001.

## 3. Results

### 3.1. Myogenic Differentiation and Mechanical Stress Trigger Muscle Cells to Produce and Release Galectin 3

With the aim of identifying the novel molecules secreted from muscle cells with a role in the interplay between muscle and bone, the culture media obtained from C2C12 cells differentiated at different time-points were subjected to a proteomic analysis. The results shown in Appendix A revealed the presence of several protein species. Some of these proteins (i.e., osteoglycin and decorin) are already known for being involved in bone remodeling [7,8]. Interestingly, another protein identified was Gal-3. Since there is evidence demonstrating that circulating Gal-3 has a role in bone physiology [21], we decided to focus our attention on this lectin and investigate its putative role as a mediator of muscle-to-bone crosstalk. To confirm our proteomic results and explore the timing of the expression of Gal-3 during muscle differentiation, the total cell lysates and the cultured media from the C2C12 cells were collected at different time-points of muscle differentiation and evaluated for the presence of Gal-3 (Figure 1A,B). The results showed that the expression of intracellular Gal-3 increased during myogenic differentiation. Intriguingly, intracellular Gal-3 up-regulation corresponded to an increase in the Gal-3 released in the culture medium at the late stage of myogenic differentiation (day 9–day 11), (Figure 1A,B). Since it is well known that physical exercise promotes bone remodeling also through the activity of several myokines that may alternatively target the proliferation and/or differentiation of bone cells [22], we asked whether Gal-3 could participate in the interplay between muscle and bone. With the aim of exploring whether the mechanical stimulation of muscle cells promoted Gal-3 up-regulation and release, differentiated C2C12 cells were exposed to mechanical strain for 6 and 24 h and were analyzed for the presence of Gal-3. The results shown in Figure 1C–E demonstrated that mechanical stimulation triggered the muscle cells to produce and release Gal-3. The lack of apoptosis identified by the western blot analysis and shown in Appendix A excludes the possibility that Gal-3 release is induced as a side effect of mechanical damage. To investigate whether mechanical stimulation triggers Gal-3 up-regulation in “in vivo” conditions, the expression levels of this lectin were analyzed in the TA and gastrocnemius muscles of mice that were subjected to training, with sedentary mice serving as the control group. Our western blot results that were obtained from the TA muscles showed that, although statistical significance was not reached, the group of trained mice exhibited higher Gal-3 expression (measured as a mean value among four animals) compared to the sedentary mice group (Figure 2A,B). Subsequently, we examined the expression of Gal-3 on the tissue sections from the TA and gastrocnemius muscles to investigate whether Gal-3 was predominantly expressed in specific subsets of fibers. Immunohistochemistry staining revealed a significantly higher level of protein positivity in the white fibers of both muscle sections, which are primarily distinguished by a fast metabolism (Figure 2C,F). This observation might elucidate why statistical significance could not be obtained in the western blot analysis, for which the Gal-3 levels of all the fibers were combined. The observed trend of increased Gal-3 expression is consistent with findings from other research groups using murine models [28,29] and further supports our results that were obtained in the muscle cell line that was exposed to mechanical stimulation (Figure 1C–E).

### 3.2. Gal-3 Secreted from Muscle Cells Reduces the Osteogenic Potential of MC-3T3 Cells

To investigate the effects elicited by the circulating Gal-3 in osteogenesis, MC-3T3 osteoblast precursors, after having proliferated (D0) and differentiated in the osteogenic medium for 3 and 7 days (D3 and D7), were treated with a recombinant soluble form of Gal-3 (rec-Gal-3) or another vehicle (Figure 3A–I). At the end of the treatment, the cells were subjected to alizarin red staining, which is generally used as a marker of bone differentiation. Figure 3A showed that supplementation with rec-Gal-3 reduced osteogenic differentiation after 3 and 7 days. Strikingly, the inhibitory effect of the rec-Gal-3 addition was counteracted by the administration of a neutralizing antibody against Gal-3 (Figure 3A,B). The addition of rec-Gal-3 to the culture medium of osteoblasts did not modify the expression of bone alkaline phosphatase (b-ALP) (Figure 3C,D), whereas it inhibited its activity, as monitored by an ALP activity assay (Figure 3E). Our findings obtained from the differentiated MC-3T3 osteoblast progenitors indicated that the supplementation of the differentiation medium with Gal-3 counteracted the expression of Osterix, which was instead promoted during the normal differentiation of these cells (Figure 3C–F). Among the signaling pathways modulated during osteogenesis, the activation of the Notch receptor is the most typical [2,3,21]. The Notch signaling pathway inhibits osteoblast differentiation [30], and since our results showed that Gal-3 impairs osteoblast differentiation, we analyzed the expression and activity of Notch (Figure 3G–I). The western blot analysis shown in Figure 3G demonstrated that the supplementation of recombinant Gal-3 into the differentiation medium of MC-3T3 osteoblast progenitors reduced the level of the expression of Notch1 during the early steps of differentiation (compare lanes “D3 and D3 + rec-Gal-3” with lanes “D7 and D7 + rec-Gal-3”). The inhibitory effect of Gal-3 on Notch1 expression was evident by the third day of differentiation, while it was restored by 7 days of differentiation. In parallel, Gal-3 was able to strongly induce cleaved Notch1 (Figure 3G,H) that, in turn, activated the transcription of Hey1 and c-Myc, two of the most known Notch1 target genes (Figure 3I). A similar trend of the modulation of the Notch level of expression has already been reported by Nakashima et al. [21,22,23,24,25,26,27,28,29,30]. In that paper, the authors suggest that Gal-3 is able to impair osteoblast differentiation by activating Notch signaling during the first days of osteogenetic induction. To confirm these considerations and further define if muscular Gal-3 has a role in the early osteogenetic processes, the supernatants of muscle cells that were overexpressing Gal-3 were next assayed for their ability to modulate the differentiation of osteoprogenitor cells. Thus, MC-3T3 osteoblast progenitors were induced to differentiate in a medium conditioned with the supernatant of Gal-3-overexpressing C2C12 cells, or, as a control, with that one from empty vector (ev)-transfected-cells. Osteoblast differentiation was hence followed by alizarin red staining. The results shown in Figure 3J,K revealed that matrix mineralization was reduced by supplementation with the supernatant from the Gal-3-overexpressing C2C12 cells, but not by the supplement added to the control sample (FLAG-Gal-3 vs. empty vector) (Appendix A). Finally, to mimic the physiological condition, the MC-3T3 osteoblast progenitors were differentiated for three days in a medium conditioned with the supernatant of the C2C12 cells that were exposed to mechanical strain for 6 h (CM Stretch). The CM medium was able to reduce ALP activity and thus osteoblast differentiation compared to the MC-3T3 osteoblast progenitors differentiated in a medium conditioned with the supernatant of the unstrained C2C12 cells (CM NO Stretch) (Figure 3L). The addition of a neutralizing antibody against Gal-3 (CM Stretch + Gal-3 Ab) was able to stimulate the ALP activity, confirming that the inhibitory effect on ALP activity is specifically due to the increased presence of Gal-3 in the culture medium, released by the C2C12 cells as a consequence of strain (Figure 3L). An RT-qPCR analysis of these samples revealed that the activation of the *RUNX2* gene, which is a marker of osteogenic processes, decreased upon supplementation with the supernatant of the strained C2C12 cells, while it was restored by the addition of anti-Gal-3 antibody to the same medium (Figure 3M).

### 3.3. Proteomic Dynamics Induced by Rec-Gal-3 during Osteogenic Differentiation

To elucidate the molecular mechanism underlying the Gal-3 dependent down-regulation of osteogenesis, the MC-3T3 cells were induced to differentiate either in the presence or absence of rec-Gal-3 and were analyzed by proteomic mass spectrometry (Figure 4A,B and Appendix A). The quantitative analysis showed that, in the presence of Gal-3, a statistically significant fold change ≥ 4 resulted for 12 up- and 1 down-regulated proteins (Figure 4B). Although no significative enrichment emerged in the molecular and biological processes related to the regulation of bone differentiation (Appendix A), the spectral counting analysis revealed the down-regulation of USP5, a Ubiquitin-Specific Protease, involved in the regulation of bone remodeling (Figure 4B) [31]. At the same time, the proteins that were up-regulated were mainly involved in basic biological processes, such as cell structure, transcription, and metabolism, confirming that Gal-3 is able to regulate fundamental cellular functions, such as proliferation, migration, differentiation, and apoptosis (Figure 4B).

### 3.4. The Analysis of the Signaling Pathways Activated by Rec-Gal-3 during Osteogenic Differentiation

Based on our mass spectrometry data, we moved to further explore the molecular mechanisms behind the effects of Gal-3 on osteoblast differentiation. To this end, we induced the MC-3T3 cells to differentiate with or without recombinant Gal-3 and evaluated the activation of the most important osteogenesis-related signaling pathways by Reverse-Phase Protein microArrays (RPPA) (Figure 5A,B). In particular, we explored the signaling cascades downstream of the RTKs, involving both the PI3K/Akt/mTOR and the STATs axes.

The heat map shown in Figure 5A demonstrated that at basal conditions, the Akt/mTOR pathway is activated during both the early (D3) and the middle (D7) phases of osteogenic differentiation. Interestingly, the supplementation of rec-Gal-3 in the differentiation medium did revert the activation of the Akt/mTOR pathway during the early phase of osteogenesis (D3), but not after 7 days of differentiation (D7).

### 3.5. Gal-3 Inhibited Bone Morphogenetic-Dependent Signaling and Counteracted the Adhesion of Osteoblasts to ECM Proteins

Since the evidence described in this study showed that both muscular and recombinant Gal-3 reduced osteoblast differentiation, the relation between soluble Gal-3 and the production of bone-specific cytokines or factors was next investigated through a bone metabolism assay (Figure 5B). Our data demonstrated that soluble Gal-3 promptly reduced, in a statistically significant manner, the level of expression of some osteogenic factors, i.e., BMP-2, -6, and -7 (D3+Gal-3 vs. D3). Collectively, the results confirmed the functional activity of extracellular Gal-3 in bone formation. Finally, to assess whether rec-Gal-3 affected the interaction between osteoblasts and bone-specific ECM proteins, we next performed an ECM adhesion assay. Briefly, the differentiated MC-3T3 osteoblast progenitors were plated on some of the most common bone-specific ECM proteins, including Collagen-I, Fibronectin or Laminin, and grown in the presence of rec Gal-3 or another vehicle for 16 h (Figure 6). At the end of the experiments, the not-attached cells were washed out, while the adherent ones were stained and counted. The results shown in Figure 6 demonstrated that soluble Gal-3 significantly decreased the adhesion ability of the differentiated cells compared to the vehicle-treated sample (D3 + rec-Gal-3 vs. D3). Altogether, these data suggested that circulating Gal-3 negatively modulated the osteogenic differentiation potential, both by inhibiting pro-osteoblastic signaling cascades and by impairing the interaction of the cells with the ECM proteins.

## 4. Discussion

Bone and skeletal muscle are mesenchymal-derived tissues whose activity, together with that of cartilage, ligaments, and tendons, is functionally and structurally interconnected. There is a multitude of reports demonstrating the existence of a crosstalk between skeletal muscle and bone, due to a number of small molecules released by muscles, the so-called myokines, able to modulate bone formation, maturation, and resorption during embryogenesis and growth. These molecules are also crucial for bone remodeling, which responds to the need for an adaptation to metabolic demands, mechanical strain, inflammation, or injury [7,8]. In this study, in an effort to identify novel putative myokines involved in the regulation of bone physiology, we discovered that Gal-3, a molecule already reported for counteracting bone differentiation, is released from muscle C2C12 cells during myogenic differentiation and mechanical strain. The positive effect of mechanical stimulation on Gal-3 expression was also observed in vivo in the muscles of trained mice [29]. We assessed the expression levels of Gal-3 in the white TA (WTA), predominantly comprising type IIB fibers, and the red TA (RTA), primarily composed of type IIA and IIX fibers [32] The analysis revealed a significant increase in Gal-3 expression in the type IIB fibers of trained mice compared to sedentary mice (Figure 2C) (*p* < 0.05). The same analysis was carried out with similar results on the gastrocnemius muscle (Figure 2D) (*p* < 0.05). Our evidence strikingly revealed that the differentiative program of MC-3T3 osteoprogenitor cells was impaired when their media were supplemented with the culture medium of the C2C12 cells that were overexpressing Gal-3 (Figure 2 H,I). Confirming previously reported evidence [21,33], the same inhibitory effect on osteoblast differentiation was also observed in the presence of a recombinant soluble form of Gal-3 (Figure 2A–D). Notably, we observed an increase in Gal-3 that can block the process of bone differentiation even after a physiological event like mechanical stretching, as Gal-3 was produced after the stretching protocol. Interestingly, this effect was abrogated following the supplementation of a Gal-3-neutralizing antibody in the culture medium. Although it cannot be excluded that an indirect action mediated by a Gal-3 effector released the culture medium of the cells, this is the first evidence of a role for Gal-3 in muscle-to-bone crosstalk. Given the possibility that the soluble recombinant form of Gal-3 may target the same osteogenic signaling pathways that are impaired by muscle-derived Gal-3, we next investigated which mechanisms were most substantially impacted. Our results demonstrated that soluble Gal-3 reduced bone differentiation by reducing the amount of Osterix (Figure 3C,D), increasing the activity of NOTCH1 (Figure 3F–H), and impairing the Akt/mTOR and STAT3 signaling pathways (Figure 5A). The last effect was observed only during the early stage of osteogenic differentiation, as rec-Gal-3 supplementation was completely ineffective on these pathways after 7 days of culture. It is not uncommon to find myokines with a precise timing of action. Osteoglycin, for example, has opposite effects on bone differentiation, depending on the phase of differentiation in which it is released [7]. Our RPPA results confirm the published results on the crucial role of STAT3 activity in bone homeostasis by regulating osteogenesis [34]. It is interesting to note that, for the first time, our results showed that soluble Gal-3 significantly decreases the expression of BMP-2,6 and 7 (Figure 5B), which are transcription factors that regulate the expression of several osteogenic factors, including, for example, Osterix. In attempts to establish a role for circulating Gal-3 in influencing the ability of osteoblasts to adhere to ECM, we interestingly found that the supplementation of differentiating cells with soluble Gal-3 reduced this innate property of osteoblasts (Figure 6). Cellular association to the ECM may be interrupted by the strong interaction settled by Gal-3 and integrins [35]. The lack of interaction with ECM proteins might also explain the reduced activation of the PI3K/PDK1/Akt axis that we observed in the osteoblasts treated with soluble Gal-3 (Figure 5A). In particular, the reduced activation of Akt might counteract the activation of the FAK-ERK1/2-RUNX2 signaling pathways, as well as of GSK-3β/β-Catenin [35,36] and, thus, be responsible for the impairment of osteogenic differentiation and matrix mineralization. These data offer further corroboration of the role of Gal-3 in antagonizing osteogenic differentiation by reducing, at the same time, the differentiation potential of osteoblasts and eventually of osteoclasts. Interestingly, the analysis of the signaling cascades activated during osteogenesis, performed by RPPA, suggested that the inhibitory effect induced by Gal-3 on those pathways at the beginning of the differentiation program is irreversible and is thus crucial for deciding the fate of the differentiation. In addition, our the results allow us to classify Galectin-3 into the myokines group that negatively regulate bone metabolism. In muscle–bone crosstalk, during physical activities, skeletal muscle can secrete hundreds of myokines to positively (e.g., irisin, IGF-1, FGF2, decorin, and IL-6) or negatively (e.g., myostatin and FGF21) regulate bone metabolism. Collectively, the findings here described may have an important therapeutic implication in the treatment of some groups of bone-related disorders, where an increase of osteogenic differentiation leads to inappropriate bone formation. This is, for example, observed in osteophyte formation (bone outgrowths, usually near joints), in osteoarthritis, or in more severe and rarer conditions, including intratendinous or intraligamentous ossification, and heterotopic ossification. In these disorders, the supplementation of Gal-3, eventually localized in the site of anomalous bone growth, may have beneficial effects, as it might mitigate the progressive abnormal formation of bone. Similarly, through the release of Gal-3 in a well-defined area, a constant increase in the activity of the muscles adjacent to the affected bones might help to counteract the progression of the ossification. 

## 5. Conclusions

In conclusion, the study here presented for the first time proposes a role for muscular Gal-3, released during muscle differentiation and mechanical stimulation, in the regulation of the early phase of osteogenesis. At the same time, our evidence has further dissected the molecular mechanisms exploited by Gal-3 to impair bone differentiation. Further investigations, possibly including the use of animal models in which Gal-3 is selectively silenced only in muscles, as well as a thorough study of the role of circulating Gal-3 on osteoclastogenesis and/or osteoclasts activity will allow the detailed characterization of this molecule and pave the road for the potential use of Gal-3 in the therapeutic treatment of a peculiar category of pathological conditions.

## Figures and Tables

**Figure 1 biomolecules-14-01243-f001:**
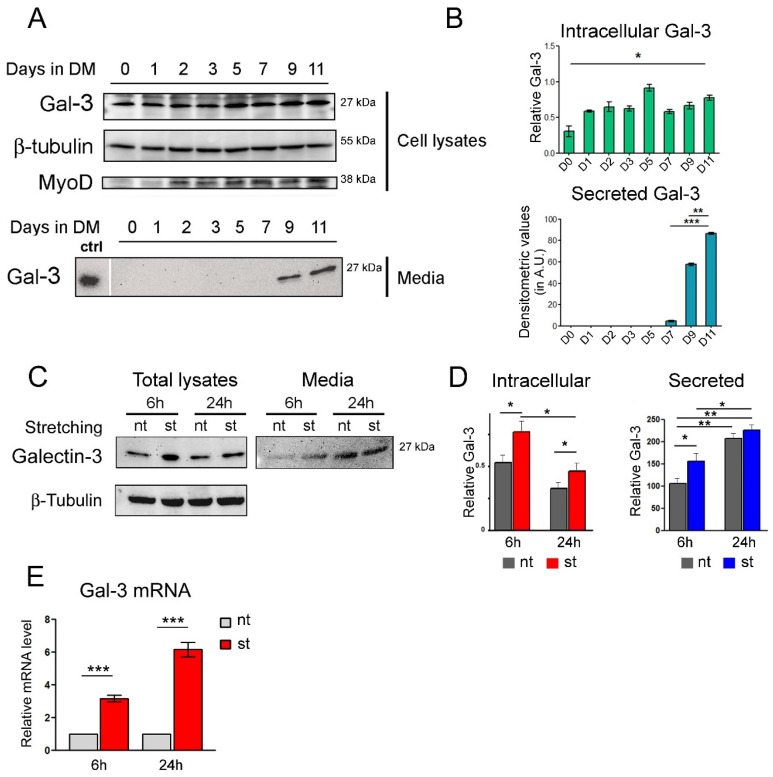
Muscle differentiation and mechanical stimulation trigger Gal-3 expression and release. (**A**) The C2C12 cells were differentiated for the indicated days. At the end of each time point, the cells and culture media were collected. The cells were lysed and, together with the culture media, were resolved by SDS-PAGE and explored for the Gal-3 level of expression. β-Tubulin and MyoD were used as loading and differentiation markers, respectively. Original images can be found in Appendix A. (**B**) The densitometric values of intracellular and secreted Gal-3 were graphed in the bar histogram, shown aside. The densitometric results were normalized on the values of corresponding β-tubulin. * *p* < 0.05, ** *p* < 0.01, *** *p* < 0.001. (**C**,**D**) The differentiated C2C12 cells were subjected to multiaxial stretching (st) for 6 and 24 h, or left untreated (nt). At the end of the stimuli, the cells were lysed and, together with corresponding media, were subjected to an immunoblot analysis to verify the Gal-3 expression. The bars are relative to three different experiments, * *p* < 0.05, ** *p* < 0.01. (**E**) The cells, as in (**C**), were lysed in Trizol and subjected to an RT-PCR analysis for monitoring the level of expression of Gal-3 mRNA. The bars show the relative amount of Gal-3 mRNA. (*n* = 3 *** *p* ≤ 0.001). Original images can be found in Appendix A.

**Figure 2 biomolecules-14-01243-f002:**
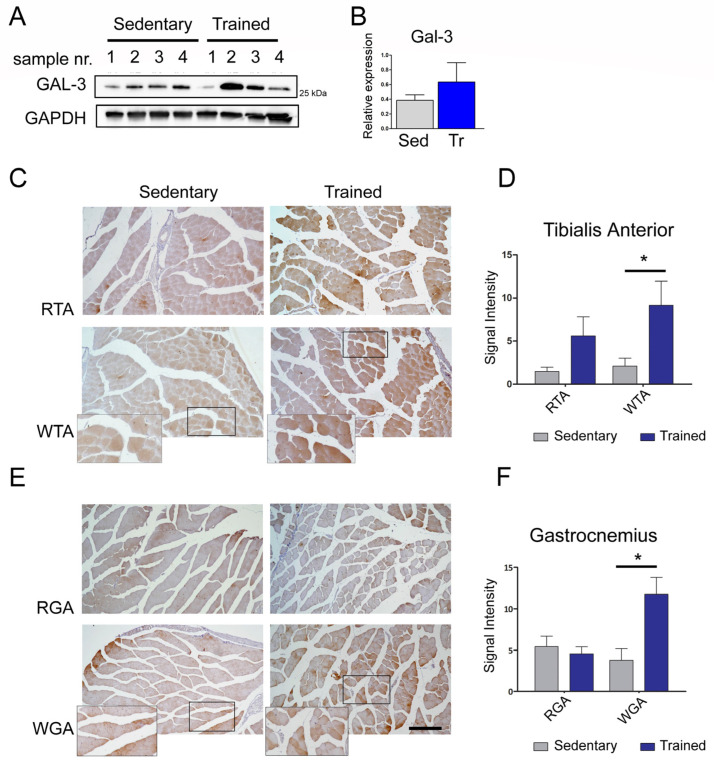
Endurance training increases Gal-3 expression in muscle fibers of type IIB. (**A**,**B**) The tibialis anterior muscles from the sedentary (Sed) and trained (Tr) mice groups were analyzed to determine the intracellular level of Gal-3 expression. The bar graph represents the average OD of the samples of the two groups, normalized to the corresponding values of GAPDH. Original images can be found in Appendix A. (**C**,**D**) A Gal-3 immunohistochemical analysis of the red (RTA) and white fibers (WTA) of the tibialis anterior muscles from the sedentary and trained mice groups with the images’ signal intensity quantification. (**E**,**F**) The Gal-3 immunohistochemical analysis of the red (RGA) and white fibers (WGA) of the gastrocnemius muscles from the sedentary and trained mice groups, with the images’ signal intensity quantification. Enlarged areas are present within the most significant panels. Bars 250 micron. A statistically significant increase in Gal-3 expression was seen in the white fibers of both muscle types in the trained mice group (*n* = 3, * *p* < 0.05), when compared to the group of sedentary mice.

**Figure 3 biomolecules-14-01243-f003:**
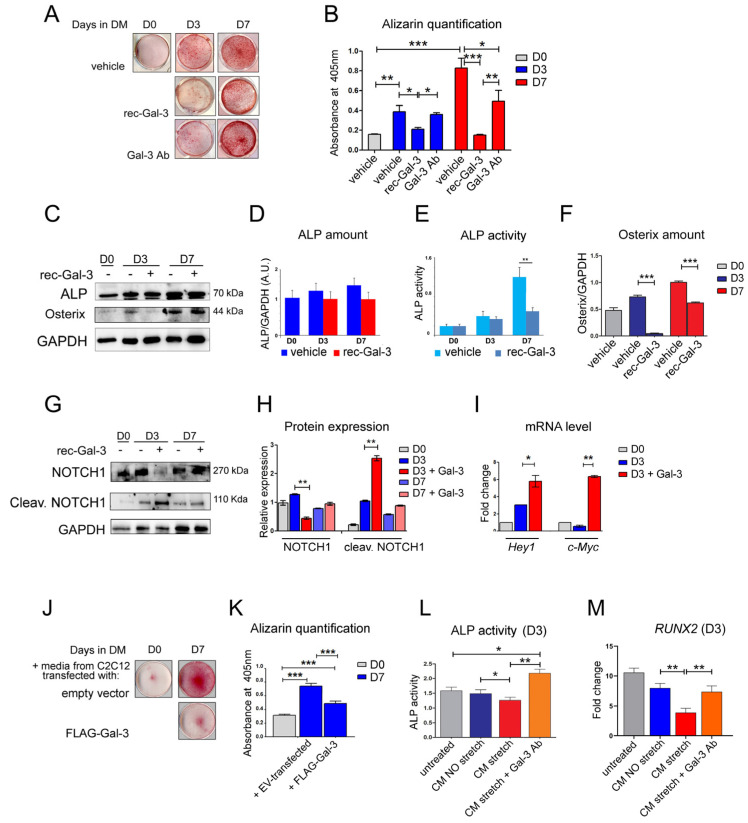
Gal-3 secreted by C2C12 cells impairs the osteogenic potential of the MC-3T3 osteoblast progenitor cells as a recombinant and as the soluble form of Gal-3. (**A**,**B**) The MC-3T3 cells were differentiated in the presence of a recombinant soluble form of Gal-3 (rec-Gal-3, FC 3 microg/mL) or another vehicle (vehicle) (*n* = 3, * *p* < 0.05, ** *p* < 0.01, *** *p* < 0.001) for three or seven days (D3 and D7, respectively). Where indicated, a combination of rec-Gal-3 and a specific antibody against Gal-3 was added (Gal-3 Ab, rabbit anti-Gal-3 1 µg/mL FC 3 µg/mL). As control, the MC-3T3 osteoblast progenitors were also collected at D0. The matrix bone mineralization was analyzed by alizarin red (AR) staining. After staining, the alizarin was diluted, quantified as described in the “Methods” Section, and the values were presented as a graph (*n* = 3, *** *p* < 0.001). (**C**) The MC-3T3 osteoblast progenitors differentiated in the presence of rec-Gal-3 or another vehicle were harvested at D0, D3, and D7 and collected. One half of the cells’ pellet was then lysed and the total lysates were resolved and assayed for bone-alkaline phosphatase (ALP) and the Osterix level of expression. GAPDH was used as equal loading control. Original images can be found in Appendix A. (**D**–**F**) The densitometric analysis of the level of expression of ALP and Osterix, normalized to GAPDH. (**E**) The other half of the cell pellets were lysed, and the ALP activity was monitored (*n* = 3, ** *p* ≤ 0.01). (**G**,**H**) The MC-3T3 osteoblast progenitors differentiated in the presence of rec-Gal-3 or another vehicle were harvested at D0, D3, and D7, lysed, and evaluated for Notch1 and cleaved Notch1 protein expression; a densitometric analysis was carried out, normalized on GAPDH levels, ** *p* ≤ 0.01. Original images can be found in Appendix A. (**I**) The graph bar of the fold change of the level of expression of *Hey1* and *c-Myc* RNAs in D0- and D3-differentiated cells (with or without rec-Gal-3), relative to three different experiments (*n* = 3, * *p* < 0.05, ** *p* ≤ 0.01). HPRT was used as a normalization housekeeping gene. (**J**,**K**) The MC-3T3 osteoblast precursor cells were maintained in the growth medium D0 and induced to differentiate in a canonical differentiation medium, D7, supplemented with the supernatant from the C2C12 cells that were overexpressing FLAG-Gal-3 (FLAG-Gal-3 OE, transfected for 48 h) or, as a control, from the C2C12 cells transfected with the empty vector (EV, transfected for 48 h). The differentiation medium (D7) was conditioned with the same amount of the C2C12 medium (EV and FLAG-Gal-3) in a medium ratio of 1:3. Samples were collected on the day of the change of the medium, D0, or at D7. The matrix bone mineralization was analyzed by alizarin red (AR) staining. The graph bars indicate the amount of AR staining (*n* = 3, *** *p* < 0.001). (**L**) The MC-3T3 osteoblast progenitors were induced to differentiate in a canonical differentiation medium, D3, supplemented with the supernatant from the C2C12 cells that were subjected or not to a stretching protocol for 6 h. The differentiation medium (D3) was conditioned with one of the three C2C12 media (C2C12 media: not stretched, stretched for 6 h, or stretched for 6 h + neutralizing antibody). The total lysates were resolved and assayed for the bone-alkaline phosphatase (ALP) level of activity (*n* = 3, * *p* < 0.05, ** *p* ≤ 0.01). (**M**) An RT-qPCR analysis of RUNX2 expression in the MC-3T3 cells differentiated as in M) (*n* = 3, ** *p* ≤ 0.01).

**Figure 4 biomolecules-14-01243-f004:**
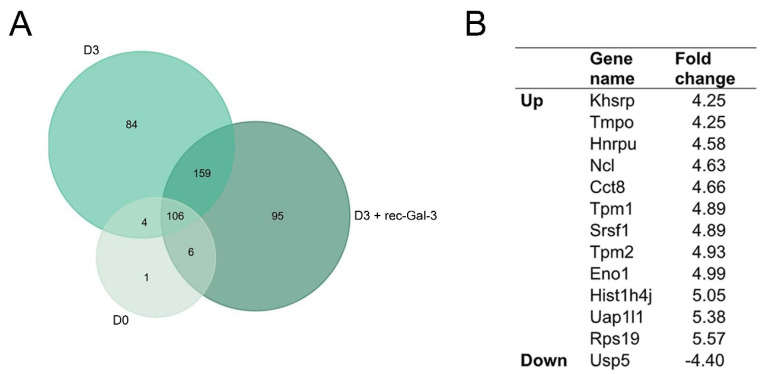
Rec-Gal-3 modulates the expression of proteins with fundamental cellular functions during the differentiation of MC-3T3 osteoprogenitor cells. (**A**) A Venn diagram showing the proteins identified by an LC-MS analysis. We identified 117 proteins on D0, 366 on D3, and 353 on D3 + rec-Gal-3. (**B**) The list of proteins up- or down-regulated in the MC-3T3 cells after 3 days of differentiation with recombinant Gal-3, compared to D3. Only the proteins identified with a fold increase ≥ 4 were reported.

**Figure 5 biomolecules-14-01243-f005:**
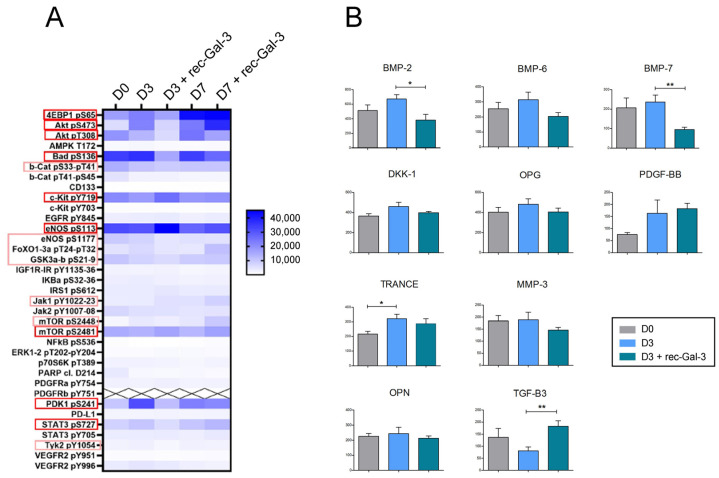
Mechanistic insights into circulating Gal-3 functions during the differentiation of MC-3T3 osteoprogenitor cells. (**A**) Heat map showing the phosphorylation of the main signaling pathways detected by the RPPA analysis during the growing state (D0), the early (D3), and the middle (D7) phases of the differentiation of the MC-3T3 cells with or without rec-Gal-3. Squared phosphorylated proteins are the species most impaired by Gal-3 supplementation. The values are reported as the mean of signal intensities, with *n* = 3. (**B**) The level of expression of the proteins promoting osteogenesis during the differentiation of the MC-3T3 cells with or without rec-Gal-3 (+rec-Gal-3) through the “Bone Metabolism Antibody Array”. The cells were blocked on D0 and D3 (*n* = 3, * *p* ≤ 0.05; ** *p* ≤ 0.01).

**Figure 6 biomolecules-14-01243-f006:**
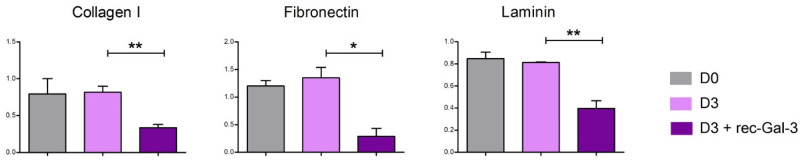
rec-Gal-3 affects the ability of differentiated osteoblasts to interact with ECM components. The evaluation of the ability of the MC-3T3 cells maintained in a growth medium (D0), differentiated for 3 days (D3) with or without recombinant Gal-3 (D3 + rec-Gal-3) to interact with the typical components of the extracellular bone matrix (Collagen I, Fibronectin, Laminin) through the “ECM adhesion assay” (*n* = 2, * *p* ≤ 0.05; ** *p* ≤ 0.01).

## Data Availability

Data are available upon request from the corresponding authors.

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
