# Peer review of "Myoblast-Derived Galectin 3 Impairs the Early Phases of Osteogenesis Affecting Notch and Akt Activity"

_biomolecules, 2024, doi:10.3390/biom14101243_

Round 1

Reviewer 1 Report

Comments and Suggestions for Authors

The manuscript titled "MYOBLAST-DERIVED GALECTIN 3 IMPAIRS THE EARLY PHASES OF OSTEOGENESIS BY AFFECTING NOTCH AND AKT ACTIVITY" by Amore et al. presents a thorough and well-executed study. The research is scientifically sound, with a clear experimental design. However, the manuscript would benefit from a more detailed exploration of the mechanism of action of Galectin-3 (Gal-3), supported by additional experiments. Furthermore, refining the abstract to better highlight the study's key findings could enhance the reader’s understanding of the significance and context of the research.

Major Points:

1. Including Alizarin Red staining images with Figure 3A is crucial. Additionally, performing ALP staining would provide stronger support for the findings in Figure 3D.

2. To further substantiate the data presented in Figures 3A, 3B, 3H, 3I, and 3L, the authors should assess other osteoblastogenic markers, such as Type I Collagen (Col1) and DMP1, using qRT-PCR or immunoblotting techniques.

3.  The introduction suggests that the soluble form of Gal-3 inhibits osteoblast differentiation by promoting osteoclast activity and reducing osteoblast differentiation (lines 92-94). To provide a more comprehensive understanding, the authors should investigate the impact of recombinant Gal-3 on osteoclastogenesis. This would clarify how Gal-3 overexpression may suppress osteoblastogenesis while simultaneously enhancing osteoclastogenesis.

4.  The results in Figure 4B should be validated with Western blot analysis to reinforce the findings.

5. The proposed mechanism of action is currently underdeveloped and requires further experimental validation. Additional studies should be conducted to draw more definitive conclusions.

Comments on the Quality of English Language

Not applicable

Author Response

Reviewer 1

The manuscript titled "MYOBLAST-DERIVED GALECTIN 3 IMPAIRS THE EARLY PHASES OF OSTEOGENESIS BY AFFECTING NOTCH AND AKT ACTIVITY" by Amore et al. presents a thorough and well-executed study. The research is scientifically sound, with a clear experimental design. However, the manuscript would benefit from a more detailed exploration of the mechanism of action of Galectin-3 (Gal-3), supported by additional experiments. Furthermore, refining the abstract to better highlight the study's key findings could enhance the reader’s understanding of the significance and context of the research.

Major Points:

Comment 1: Including Alizarin Red staining images with Figure 3A is crucial. Additionally, performing ALP staining would provide stronger support for the findings in Figure 3D.

Reply to Comment 1: We thank the reviewer for this suggestion. Alizarin images corresponding to the histogram bar of Figure 3A have been added to the new version of the manuscript.

Regarding ALP, we totally agree with the Reviewer that ALP staining would be useful to visualize the protein in its biological context, but we believe that being not relative to its activity, it would not give additional information compared to those already shown. Western Blot shown in Figure 3B (now 3C) allowed us to make a qualitative and quantitative assessment of the protein, whereas the assay of alkaline phosphatase activity of Figure 3D (now 3E) helped us to quantify the activity of this enzyme.

Comment 2: To further substantiate the data presented in Figures 3A, 3B, 3H, 3I, and 3L, the authors should assess other osteoblastogenic markers, such as Type I Collagen (Col1) and DMP1, using qRT-PCR or immunoblotting techniques.

Reply to Comment 2: We are sorry for having generated a certain confusion. Figures 3A, 3B, 3H, 3I, and 3L demonstrate that commercially derived recombinant Galectin-3, or Galectin-3 released from muscle cells following their transfection with the corresponding cDNA, or released from muscle cells upon strain, induce a reduction of osteogenic differentiation. This observation has been verified by both Alizarin Red staining, alkaline phosphatase activity and Notch cleavage analysis. We thank the reviewer for the helpful suggestion to evaluate the modulation of other osteogenic markers by a biochemical and molecular approach. We decided to follow the modulation of Osterix (presented in the new Figure 3C), which is an early differentiation marker for osteoblasts and osteocytes, as well as the activation of RUNX2 gene (new Figure 3N), which encodes a transcriptional factor regulating osteogenic-specific genes (including also Osterix).

Comment 3:  The introduction suggests that the soluble form of Gal-3 inhibits osteoblast differentiation by promoting osteoclast activity and reducing osteoblast differentiation (lines 92-94). To provide a more comprehensive understanding, the authors should investigate the impact of recombinant Gal-3 on osteoclastogenesis. This would clarify how Gal-3 overexpression may suppress osteoblastogenesis while simultaneously enhancing osteoclastogenesis.

Reply to Comment 3: We thank the reviewer for this comment, and once again we apologize for not being sufficiently clear. Lines 92-94 to which the reviewer refers, illustrate indeed the “state of the art” of published evidence of Galectin 3 on bone metabolism. However, in this paper, as also it is suggested in its title, we decided to focus on the effects of Galectin 3 on osteogenic differentiation only.

Comment 4.  The results in Figure 4B should be validated with Western blot analysis to reinforce the findings

Reply to Comment 4: We appreciate the Reviewer’s suggestion to validate our label-free mass spectrometry results by western blotting. While western blotting can be indeed a valuable tool, we believe that our MS data, in the context of the proposed work, provide a comprehensive and robust analysis of the proteomic changes associated with Gal-3. Moreover, in the table presented in Figure 4, we report only proteins up/down-regulated with a cut off value of 4-fold changes, to ensure the accuracy and reliability of our results. Since the scope of our proteomic study was to elucidate the targets influenced by the administration of rec-Gal-3 during the osteogenic-induced differentiation, we feel that the observed changes in protein abundance could provide valuable insights into the role of Gal-3 and being of interest for future studies of our and other research groups.

While we understand the Reviewer’s desire for additional validation, we would like to emphasize that in our case, validation of proteomic data should not be limited to the sole evaluation of protein expression modulation but, more importantly, should be extended to the identification of the role these proteins play in osteogenesis within the context of our model.

In this sense, we believe that given the complexity of the rigorous experimental design this research deserves, such required validations might warrant a dedicated study.

Comment 5. The proposed mechanism of action is currently underdeveloped and requires further experimental validation. Additional studies should be conducted to draw more definitive conclusions.

Reply to Comment 5: We thank the reviewer for her/his valuable consideration. We firmly believe that all the results shown in our study (modulation of strategic transcription factors, or signaling pathways, impairment of mineralization or ALP activity, modulation of the attachment affinity to ECM substrates etc) converge to demonstrate that muscle cells secrete and release Galectin 3, which in turn attenuates bone differentiation. Nonetheless, these conclusions have also been corroborated by selective inactivation of Galectin 3 by the use of anti-galectin 3 antibodies. It is the first study assessing this role for muscle-released Galectin-3. However, at the same time, we agree with the reviewer that this research needs additional evaluations. To further validate our thesis, we are currently developing inducible mice models in which the expression of muscular Galectin 3 may be specifically ablated. This approach will ultimately assess the true function of muscle-derived Galectin 3 in bone homeostasis and remodeling also in in vivo models.

Reviewer 2 Report

Comments and Suggestions for Authors

In this manuscript from Emanuela Amore and colleagues, the authors sought to identify molecules secreted from skeletal muscle that impact bone physiology. They identified galectin (gal3) and performed a series of experiments that show gal3 produced by muscle can inhibit osteoblast differentiation. Overall this is a very nice paper. The experiments appear to have been well designed, the figures present their data effectively, conclusions are reasonable interpretation of their data and the writing is clear and concise.

I really only have two suggestions for improvement, neither of which represent significant concerns:

1.        In figure 3 they present alizarin red staining data as absorbance units. However, in the Methods they only discuss taking photos of the AZR staining and presenting representative images (lines 180-183), not any quantitative assay. Please revise to bring the Methods and figure data into accord.

2.        I’m not deeply immersed in the muscle-bone crosstalk literature. Most of what I’m familiar with are findings that muscle activity promotes bone formation, so their overall finding of muscle-derived antiosteogenic factor is a little counter-intuitive to me. Perhaps they might add a few sentences around their thoughts on this point to the discussion. What might the normal physiological role for this effect in vivo?

Author Response

Reviewer 2

In this manuscript from Emanuela Amore and colleagues, the authors sought to identify molecules secreted from skeletal muscle that impact bone physiology. They identified galectin (gal3) and performed a series of experiments that show gal3 produced by muscle can inhibit osteoblast differentiation. Overall this is a very nice paper. The experiments appear to have been well designed, the figures present their data effectively, conclusions are reasonable interpretation of their data and the writing is clear and concise.

I really only have two suggestions for improvement, neither of which represent significant concerns:

Comment 1.        In figure 3 they present alizarin red staining data as absorbance units. However, in the Methods they only discuss taking photos of the AZR staining and presenting representative images (lines 180-183), not any quantitative assay. Please revise to bring the Methods and figure data into accord.

Reply to Comment 1: We are grateful to Reviewer 2 for well rating our study, and we apologize for generating confusion. In the revised version, we added the experimental procedure of quantitative analysis of AZR in the “Methods” section. Moreover, Methods and Figure 3 have been brought into accord.

Comment 2.        I’m not deeply immersed in the muscle-bone crosstalk literature. Most of what I’m familiar with are findings that muscle activity promotes bone formation, so their overall finding of muscle-derived antiosteogenic factor is a little counter-intuitive to me. Perhaps they might add a few sentences around their thoughts on this point to the discussion. What might the normal physiological role for this effect in vivo?

Reply to Comment 2: Mechanical communication between bone and muscle is associated to biochemical inputs generated by the production and secretion of soluble factors by muscle and bone which act towards bone and muscle. We agree with the Reviewer and to the general concept that muscle activity promotes bone formation, but we want to point the attention onto particular classes of myokines that go “against the current” and counteract osteogenesis. This is for example the case of myostatin which, together with irisin and IL-6 represent a good model of bone homeostasis regulated by time and muscle activity. Physical activity stimulates the production and secretion of irisin, IL-6 and myostatin. But myostatin and irisin have an opposite action in bone–muscle metabolism: physical activity primarily stimulates the release of myostatin, which is part of the TGFβ-superfamily, that is a positive regulator of bone resorption and reduce the formation of new bone inhibiting the differentiation of mesenchymal stem cells into osteoblasts; subsequently, physical activity promotes the release of irisin which stimulates osteogenic and skeletal differentiation, inhibiting TGFβ-signaling and the expression of sclerostin.

Similarly, Galectin 3 release following muscular activity might counteract the formation of new bone of nearby districts. This action might be transitory and for example terminate following the release of other myokines featuring synergistic or opposite effects on bone homeostasis. Our unexpected finding that under training, Galectin 3 is mainly produced by fast twitch fibers might indicate a refined mechanism of action activated only under particular conditions that deserve further investigation.

Round 2

Reviewer 1 Report

Comments and Suggestions for Authors

In the future, the proposed mechanism of action should be further studied to reach more definitive conclusions.